# Additive Manufacturing of an Extended-Release Tablet of Tacrolimus

**DOI:** 10.3390/ma16144927

**Published:** 2023-07-10

**Authors:** Azin Abdollahi, Zahra Ansari, Mohammad Akrami, Ismaeil Haririan, Simin Dashti-Khavidaki, Mohammad Irani, Mojtaba Kamankesh, Emad Ghobadi

**Affiliations:** 1School of Pharmacy, International Campus, Tehran University of Medical Sciences, Tehran 1416634793, Iran; axin.abdollahi@gmail.com; 2Department of Surgery and Radiology, Faculty of Veterinary Medicine, University of Tehran, Tehran 1416634793, Iran; zahraansariii94@gmail.com; 3Department of Pharmaceutical Biomaterials, Medical Biomaterials Research Center, Faculty of Pharmacy, Tehran University of Medical Sciences, Tehran 1416634793, Iran; haririan@tums.ac.ir (I.H.); emad.q47@gmail.com (E.G.); 4Institute of Biomaterials, University of Tehran and Tehran University of Medical Sciences (IBUTUMS), Tehran 1416634793, Iran; 5Department of Pharmaceutics, Faculty of Pharmacy, Tehran University of Medical Sciences, Tehran 1416634793, Iran; 6Department of Clinical Pharmacy, Faculty of Pharmacy, Tehran University of Medical Sciences, Tehran 1416634793, Iran; dastis@sina.tums.ac.ir; 7Faculty of Pharmacy, Alborz University of Medical Sciences, Karaj 56131452, Iran; m.irani@abzums.ac.ir; 8Department of Polymer Chemistry, School of Chemistry, College of Science, University of Tehran, Tehran 1416634793, Iran; kamankesh@ut.ac.ir

**Keywords:** 3D printing, additive manufacturing, tacrolimus, tablet, extended release, personalized medicine

## Abstract

An extended-release tablet of tacrolimus as once-daily dosing was fabricated using 3D printing technology. It was developed by combining two 3D-printing methods in parallel. Indeed, an optimized mixture of PVA, sorbitol, and magnesium stearate as a shell compartment was printed through a hot-melt extrusion (HME) nozzle while an HPMC gel mixture of the drug in the core compartment was printed by a pressure-assisted micro-syringe (PAM). A 3D-printed tablet with an infill of 90% was selected as an optimized formula upon the desired dissolution profile, releasing 86% of the drug at 12 h, similar to the commercial one. The weight variation, friability, hardness, assay, and content uniformity determination met USP requirements. A microbial evaluation showed that the 3D-printed tablet does not support microbial growth. SEM analysis showed smooth surfaces with multiple deposited layers. No peak interference appeared based on FTIR analysis. No decomposition of the polymer and drug was observed in the printing temperature, and no change in tacrolimus crystallinity was detected based on TGA and DSC analyses, respectively. The novel, sTable 3D-printed tablet, fabricated using controllable additive manufacturing, can quickly provide tailored dosing with specific kinetic release for personalized medicine at the point-of-care.

## 1. Introduction

Transplantation has been defined as the relocation of healthy tissues or organs to different parts of the recipient’s body, restoring the function of the faulty organ in the recipient. According to the recent WHO reports on approximately 80.4% of the world’s population, there have been roughly 12,981 solid organ transplants each year, comprising 31.7% kidney transplants, 20.1% liver transplants, and the remaining being related to heart, lung, pancreas and S, bowel transplantations [1]. Among them, non-autograft transplantations elicit an immune response, sometimes followed by rejection, requiring immunosuppressant medicines [2].

Immunosuppressant medication has been used to prevent/treat graft rejection and graft-versus-host diseases and has been considered to reduce tissue damage, which occurs in autoimmune or inflammatory reactions. The immunological response has been modulated via the following major drug classes: corticosteroids such as methylprednisolone and dexamethasone; cytotoxic suppressants such as chlorambucil, cyclophosphamide, mercaptopurine, and methotrexate; suppressant antibodies such as infliximab and basiliximab; mTOR-inhibitor derivatives of sirolimus such as everolimus; and finally, calcineurin inhibitors such as cyclosporine and tacrolimus [3].

Tacrolimus is known as a macrolide inhibitor of calcineurin which was first isolated from bacterium streptomyces tsukubaensis in 1987 [4]. Tacrolimus, under the brand name of Prograf^©^, was commercially made for kidney and liver transplantations in 1994 [5]; however, due to its good performance, it is currently used in heart, lung, and bone marrow transplantations as the first line of therapy in conjunction with corticosteroids and drugs such as mycophenolate mofetil [6]. Evidence has shown that it binds to FK506 binding protein (FKBBP12), thus creating a complex that inhibits calcineurin, ultimately reducing lymphocytes’ signal transduction and IL-2 transcription [7]. A more favorable lipid profile has also been reported for tacrolimus administration, elongating its duration of action; however, the incidence of acute graft rejection is lower than that of sirolimus [5]. Tacrolimus is usually taken twice per day and may cause nausea, irregularities in stomach function, and dose-dependent sebaceous hyperplasia [8]. In addition, the drug is taken with food, thus reducing its absorption.

Controlled-release drug delivery systems for tacrolimus are reported on liposomal forms, nanoporous materials, electromechanics, and hydrogel-based and polymeric systems [9,10,11,12]. Liposomal systems have been extensively utilized, yet in vivo instability and their removal by the reticuloendothelial system are the most significant challenges [13]. The utilization of pump devices to release the amount of the drug needed is also expensive and carries the risk of dose-dumping in case of an error. In recent years, oral controlled-release technologies have improved adherence, reducing the hassle of repeated drug administrations [14].

Tacrolimus, in extended-release forms, has already become available under the brand names Astagraf^©^ (Tokyo, Japan) and Advagraf^©^ (Toronto, ON, Canada). These forms have shown better patient compliance with a reduced dosage of once per day, stable systemic absorption, and efficacy and safety comparable to common forms in patients with renal and liver transplants.

Despite the improved efficacy in the extended-release form, concerns of toxicity or graft rejection related to immunosuppressant drugs’ pharmacologic and metabolic profiles must be considered for treatment. The variable bioavailability of tacrolimus, which is reported between 5% to 67% and is due to the first-pass effect, can put the patients at the risk of graft rejection. Furthermore, pharmacokinetic differences due to age, race, illnesses, and dissatisfaction with treatment lead to poor clinical consequences. For example, low absorption has been reported in children for tacrolimus FK506 due to increased removal by the liver and kidney. Furthermore, diabetic patients have shown reduced absorption for FK506, hence needing more drugs than non-diabetic patients; therefore, transplant recipients must undergo therapeutic dose monitoring to improve clinical efficacy and reduce toxicity caused by therapy [15].

Personalized medicine refers patients to particular medications according to their individual characteristics including their genetic profile, phenotypic response, and their pathology. New technologies should be introduced to produce standard dosage forms rapidly at the point-of-care. The production of personalized tablets has recently been facilitated by 3D-printing techniques and additive manufacturing [16,17]. Compared to other manufacturing methods, this technology is known as very efficient and economical. 3D printing, by enabling spatial distribution of the drug-polymer combination, is considered as an alternative to engineering and specifying the release profile of drugs. Spirtam^©^ (Mason, OH, USA), as the first 3D-printed drug approved by the FDA, is an example from such a method [18].

The most notable benefits of the approach are mentioned: (1) Stability of multiple drugs in a single tablet; (2) Producing an exact dose of each drug according to the pharmacokinetic profile; (3) Controlling the release kinetics of each drug in designed structure by balancing the dissolution profile [19].

In pharmaceutics, hot-melt extrusion (HME) has been used to improve solubility, create a controlled release, and, in some cases, cover the harsh taste of the drugs [20]. The HME-based 3D-printing method in parallel with another extrusion method such as pressure-assisted microsyringe (PAM) printing have made it possible to develop 3D-printed tablets [21,22].

Here, we have formulated an extended-release tacrolimus tablet using extrusion-based 3D-printing techniques for the first time. The polymer mixture for the shell and drug-containing gel for the core compartment of printer inks were optimized to obtain the desired release profile. The 3D-printed tablet was then physiochemically characterized in terms of API and excipient interaction, decomposition, in vitro drug-release behavior, microorganism contamination, stability, and other USP requirements.

## 2. Material and Methods

### 2.1. Materials

Tacrolimus monohydrate was bought from sigma company (Darmstadt, Germany). PVA filaments were purchased from Ultimaker company (Utrecht, The Netherlands). Magnesium stearate and sorbitol were purchased from Merck CO. (Darmstadt, Germany). Hypromellose was provided by Samsung Fine Chemicals Co. (Seoul, South Korea). All solvents were supplied from Merck CO.

### 2.2. Tablet Design and Filament Preparation

The pattern of the tablet form, with a round shape of about 12 mm in diameter and 5 mm in height, was designed with SolidWorks software (v. 2022, Dassault Systems, Vélizy-Villacoublay, France). The dimension of the design was further modified based on experimental work. For the shell compartment, the various ratios of PVA filaments, magnesium stearate, and sorbitol were mixed and melted for 20 min to get an optimized ratio based on extrudability and printability in a hot-melt extruder. For the inner gel, different ratios of tacrolimus, ethanol, and HPMC were mixed. The best ratio based on extrudability and printability was chosen for printing using a second extruder, PAM. Finally, the filament and gel were introduced into the 3D printer (3D BIO, Omid Afarinan Co., Tehran, Iran).

### 2.3. Preparation of 3D-Printed Tablets

The tablet layout was uploaded by Repetier slicing software (version 2.2.4) and then printed with 3D Bio using two independent extruders. First, PVA filament was extruded and printed using working stepper motors which directed filaments into the hot-melt container equipped with a 0.4 mm nozzle. The printing process was conducted at 150–225 °C upon formulations at 10 mm/s speed, a bed temperature of 50 °C, and a layer height of 0.4 mm. In parallel, the second PAM conducted the drug-containing gel into the interval space of printed PVA shells in the core layer position.

### 2.4. Morphology of Filament and Tablets

Tablet morphology was investigated using scanning electron microscopy (SEM) (TESCAN MIRA3, Brno, Czech Republic) at 10 kv and 54× to determine the dimensions of the printed layers of tablets. Before imaging, the sample was prepared on metallic stubs and gold plated in a vacuum chamber using JFC-1200 Fine Coater (Jeol, Tokyo, Japan).

### 2.5. Fourier-Transform Infrared Spectroscopy (FTIR)

The drug substance, excipient, and grinded powder from the printed tablets were mixed with KBR and then pressed separately to obtain disks. Each disk was placed in a sample position to scan the infrared spectrum in the range of 400 to 4000 cm^−1^ using infrared spectroscopy (PerkinElmer, lambda 365, Waltham, MA, USA).

### 2.6. Differential Scanning Calirometry (DSC) and Thermogravimetric Analysis (TGA)

A thermal analysis instrument (SDTQ600, V20.9 Build 20, New Castle, DE, USA) was utilized to investigate the thermal behavior of the material. Briefly, 5 mg of each sample was accurately weighed and placed in the Alu pan of the device. Then, samples were heated from 25 to 600 °C at a rate of 10 °C under a nitrogen medium to measure them in triplicate.

### 2.7. Assay, Content Uniformity, and Dissolution Studies

All measurements were performed according to the USP 43 monograph for tacrolimus determination in the dosage form. To prepare the sample solution for assay, 20 tablets were mixed, accurately weighed into amounts of milled powder equivalent to the mean weight of tablets, transferred to a 250 mL measuring flask, dissolved, shaken, and made up to volume. For the content uniformity assessment, ten printed tablets were placed separately in 250 mL flasks to dissolve in the mobile phase in a procedure similar to the assay.

For quantification, a standard stock solution of 200 μg/mL of tacrolimus was prepared and then diluted to standard solution in mobile phase with an equivalent concentration to the sample solution.

For dissolution, samples were taken from 900 mL media (0.005% HPC in water with 0.50% SLS adjusted to pH 4.5) in time intervals of 0.5, 1, 2.5, 4.5, 6.5, 8.5, and 12 h using a dissolution tester (Erweka) in which the temperature and paddle rotation had been set at 37 °C and 100 rpm, respectively.

To estimate the amount of tacrolimus in each sample, standard solutions were prepared from a common stock solution of assay after diluting it in dissolution media.

A portion of each sample solution passed through a filter of 0.45 μM before the volume of 20 μl was injected into the HPLC.

The chromatographic condition was adjusted to the mobile phase of 6% phosphoric acid; deionized water: methanol: acetonitrile with ratios of 0.1:36: 18: 46; column: 4.6 mm × 10 cm; 5 μm packing L1 (Agilent technologies, Santa Clara, CA, USA); detection wavelength of 210 nm; column temperature and flow rate were set at 50 °C and 1 mL/min, respectively. Chromatograms were taken for 25 min. The drug percentage was calculated according to the peak area of the tacrolimus in injected samples in comparison to that of the injected standards.

### 2.8. Hardness and Friability Measurement

The hardness and friability of tablets were measured using a hardness tester (Erweka, TBH 125 Series, Langen, Germany) and friability tester (Erweka TA 100, Gelderland, The Netherlands), respectively. Furthermore, weight variation was calculated using a balance (Sartorius, Goettingen, Germany) after weight measurement.

### 2.9. Microbial Tests

The streak plate method was used to determine the total microbial content using agar plates. Briefly, 1 gr of each sample was dispersed in sterile normal saline. An appropriate volume of sterile culture media of Mueller Hinton Agar and soybean casein digest agar were transferred into the plate and incubated for 24–48 h at 35 °C. Furthermore, to determine mold contamination, samples were cultured in sabro dextrose agar medium and incubated for 5–7 days at 25 °C. Finally, a differential diagnosis of *E. coli*, *Salmonella* species, *Pseudomonas aeroginosa*, *Staphylococcus aureous*, *Candida albicans*, and *Aspergillus niger* was examined according to US pharmacopeia <61> as described “Microbiological Examination of Nonsterile Products: Microbial Enumeration”.

### 2.10. Stability Study

The stability study of the 3D-printed tablets in a blister pack was carried out at 40 °C ± 2 °C/75 ± 5% RH for a duration of 36 months in a stability chamber, as recommended in ICH quality topics Q1.

## 3. Results

### 3.1. Results of Filaments Formulation and 3D Printing Process

Due to concerns about the thermal instability of the tacrolimus during hot-melt extrusion, for the 3D printing of the tablet, PAM was also utilized in parallel to the hot-melt extruder. The layout of the tablet shell is shown in Appendix A.

To fabricate the PVA filament for the shell part of the tablet, different ratios of polymer, plasticizer, and lubricant were mixed and melted to achieve extrudability and lower temperature printability, as shown in Table 1.

In addition, magnesium stearate was added to the formulation to improve extrudability. As shown, the addition of plasticizers of PEG35000 and PEG10000 did not result in printability and extrudability; however, the addition of sorbitol as the plasticizer at a considerable amount not only resolved the process of extrusion and printing, but also facilitated printing at a lower temperature. Different types of hot-melt extruded filaments upon optimization are represented in Appendix A.

As shown in Table 1, F9 was selected for further study. To print tacrolimus in a gel matrix, the best ratio of HPMC in ethanol (F2) was selected for printing through PAM as a second nozzle (Appendix A).

The sliced version by software and the image of the final 3D-printed tablet is shown in Figure 1A. Furthermore, cross-sections of different infill percentages of the printed tablet are shown in Figure 1C–E.

### 3.2. Results of SEM Analysis

The morphology of the printed filament was investigated by SEM imaging (Figure 1B). The image clearly shows that the surfaces are smooth with multiple deposited layers. However, due to the printing of the polymers being in high temperatures as well as rapid evaporation of the water content, small empty spaces exist in some sections. Furthermore, the merging of layers is visible in some areas. The high precision and accuracy of the 3D-printing process upon the thickness of the filament were confirmed using SEM with the selected nozzle diameter, which was 0.4 mm.

### 3.3. Results of FTIR Analysis

The FTIR spectrum of tacrolimus, according to Figure 2A, reveals etheric bonds (C-O-C) stretching at 1090.8 and 1173.19 cm^−1^. Furthermore, stretching vibrations of C-O at 1194.05 cm^−1^ for ester and C=O stretching at 1741.63, 1724, and 1694.29 cm^−1^ for ester and ketone bands were observed, respectively. Furthermore, the vibration of the O-H band at 3460 cm^−1^ indicated tacrolimus identification. Additionally, the 2925 and 2981 cm^−1^ bands are attributed to vibrations of symmetric CH2 and asymmetric C-H bands, respectively.

The FTIR spectrum of PVA filament showed a typical stretching band at 3416.55 cm^−1^ for hydroxyl, 2925 cm^−1^ for vibrations of C-H and 1636 cm^−1^ band for C=C vibration, as well as 1111 cm^−1^ and 1384 cm^−1^ C-H for stretching vibrations of C-O bands (Figure 2A).

According to the FTIR spectrum of sorbitol, the peaks that appear at 1047.90 and 1095.98 cm^−1^ are attributed to C-O stretching vibrations. The wavelengths at 886 and 1417 cm^−1^ are related to bending vibrations of inward and outward O-H bonds, respectively. The broad peak of O-H stretching was observed at 3341 cm^−1^.

According to the FTIR spectrum of the HPMC polymer, stretch vibrations on C-O bonds in the hexagonal part resulted in the appearance of peak at 1637 cm^−1^. The appearance of the absorption band at 3420 cm^−1^ is related to the hydroxyl polymer group of the compound, while the appearance of the 2923 cm^−1^ peak denotes the stretching vibrations of the C-H bonds. Peaks at 1117 and 1384 cm^−1^ are attributed to C-O stretching and C-H bending, respectively. The FTIR spectrum of the 3D-printed tablet showed the stretching band at 3415 cm^−1^ for the hydroxyl groups of PVA and sorbitol as well as the peak at 2924 cm^−1^ for stretching vibrations of -CH_2_. The intermediate bands at 618 and 1636 cm^−1^ overlapped with the PVA filament spectrum, while the 1117 cm^−1^ peak is attributed to C-O band vibration, matching the PVA filaments and sorbitol. Generally, all peaks were consistent with the ingredients (Figure 2A).

Furthermore, the FTIR spectrum of the physical mixture of the ingredients shows a broad peak at 3342 cm^−1^ for hydroxyl stretching and peaks at 2918 cm^−1^ for vibrations of C-H, 1095 cm^−1^ and 1048 cm^−1^ for the stretching vibrations of C-O bands, and 887 cm^−1^ and 1417 cm^−1^ for the bending vibrations of inward and outward O-H bonds. Peaks with a weak intensity at 1638 cm^−1^ and 1735 cm^−1^ are attributed to C=O stretching in ester and ketone bands of tacrolimus.

### 3.4. Results of DSC Thermal Analysis

The first endothermic peak in DSC thermal analysis of tacrolimus monohydrate occurred around 127 °C, resulting from tacrolimus melting (Appendix A). By analyzing DSC thermograms for the components of our designed pill, no significant difference was seen before 200 °C. Nevertheless, a small endothermic peak was observed at 127 °C in the thermogram of the 3D tablet, indicating no change in the drug’s crystalline structure during the printing process (Figure 2B).

### 3.5. Results of TGA Thermal Analysis

According to the TGA analysis of tacrolimus, as shown in Figure 2C, the first weight loss of about 2.5% within a temperature range below 205 °C was due to the evaporation of water content. Afterward, weight reduction in higher temperatures was due to tacrolimus decomposition. Furthermore, thermogram of TGA analysis for 3D-printed tacrolimus tablet (Figure 2C) showed weight loss of 12.7% at 230° due to evaporation of water content, which was followed by a second weight loss of about 82.5% up to 375° due to the decomposition of organic materials (polymers, drugs, etc.). Therefore, no thermal decomposition occurred in printing temperatures lower than 250 °C.

### 3.6. Results of Drug-Release Profiles, Assay, and Content Uniformity

To tune the release profile, tablets were printed with different infill percentages. After the initial evaluation, tablets with an infill of 90% were selected as extended-release tablets. The assay determination of a 3D-printed tablet of tacrolimus with infill of 90% obtained 98.9%, which was in the range of the USP monograph (93–105%) as shown in Appendix A. The relative standard deviation (RSD) for all injections was under 2%. A representative chromatogram of injected tacrolimus samples for assay determination is shown in Figure 3A.

Additionally, the results of content uniformity of the tablets met the USP requirement, as the acceptance value was below 15 (about 9.12, Appendix A). The amount of tacrolimus in each dissolution sample taken from dissolution vessels at different time intervals was calculated against the calibration curve of tacrolimus standards (Appendix A). As seen, the standard curve is entirely linear (R2 > 0.99). Photos of the dissolution trend for 3D-printed tablets are presented in Appendix A. Furthermore, according to the HPLC chromatograms obtained for the tacrolimus dissolution test, the retention time (R_t_) was around 19.5 min. The dissolution profile of the extended-release 3D-printed tablet of tacrolimus is shown in Figure 3B. A tacrolimus release of 90% occurred at 10 h for the 3D-printed tablets compared to 80 min for the commercial tacrolimus capsules. Hence, the 3D-printed tablet showed a prolonged release profile similar to the dissolution profile of the XR tablet brand. Furthermore, a less sustained release trend appeared for infills of 10% in comparison with those of 50% and 90% (Appendix A).

In the following, the physicochemical characterization of the weight variation, friability, and hardness of the tablet are investigated. As the median weight of the pills was 585 mg, according to the USP requirement, the weight variation percentages for pills above 324 mg are approved when the RSD is less than 5. The RSD of the weight variation for the 3D-printed tablets was 2.43, in the acceptable range (Appendix A). As shown in Appendix A, the friability of 3D-printed tablets, determined after comparing the initial and final weights of the pills, was in the acceptance range (about 0.12%, less than 1%). Finally, the hardness measurement of tablets was above the measuring capacity of our apparatus (above 50 kg).

### 3.7. Results of Microbial Tests

The total count which was determined by microbial tests for the 3D-printed tablet revealed that the colony-forming unit (CFU) is within the limit, and the microbial contamination was not significant. Furthermore, a low level of fungi contamination was detected in the same sample (Table 2). Furthermore, no contamination was detected for all standard bacterial strains (Table 2).

### 3.8. Results of Stability Assessment

The specifications for the 3D-printed tablets of ticagrelor at the end of their shelf life are demonstrated in Table 2. According to the stability assessment in accelerated conditions, the 3D-printed tablets of tacrolimus were considered stable after 6 months upon physiochemical tests and evaluation of micro-organism contamination.

## 4. Discussion

One of the most critical objectives in developing pharmaceutical dosage forms is maintaining the plasma concentration of the drugs and extending the duration of their action. The formulation of such a DDS with an extended-release trend not only improves patient compliance, but also minimizes fluctuations in the plasma concentration profile [23].

For decades, different strategies have been used for developing extended-release tablets as a dosage form. For example, matrix-based tablets have been formulated by pharmaceutical companies to create controlled-release forms. However, sometimes, new approaches are needed to enable rapid small-batch production of a tablet with the desired dose and specific kinetic release and meet pharmaceutical economics and quality.

3D-printing technology has offered new hopes to achieve this goal because the desired control of patient-tailored tablets and their release kinetics is possible through manipulating the polymer ratio in printer inks [16]. Taking advantage of 3D-printing technology as well as the development of extended-release dosage forms, we decided to design a 3D-printed tacrolimus tablet for the first time using HME-based 3D-printing technique.

In our study, PVA as the primary polymer and commercially available water-soluble filament were used for HME-based 3D printing [15]. To facilitate the extrudability of PVA, magnesium stearate was added to the formulation. However, the addition of the compound decreased the drug release rate due to its inherent hydrophobic property. Additionally, sorbitol appeared as the best plasticizer among the applied plasticizers due to its improvement of the extrusion and printing process at a lower temperature than the PVA filament melting point. This can be attributed to the molecular interactions of sorbitol with PVA, destroying the intramolecular interaction of PVA hydroxyl groups which influence HME behavior [24]. However, the inevitable use of higher proportions of PVA in the shell compartment to print sustained-release tablets was one of the limitations of the study. Furthermore, using a single hot-melt extruder in the fabrication of the tablet resulted in drug decomposition. Therefore, the core compartment was fabricated using an alternative PAM.

No drug and polymer interference appeared, as the FTIR spectrum of the 3D-printed tablet corresponded to that of all ingredients. However, the characteristic peaks of the tacrolimus in the FTIR spectrum of the physical drug mixtures with PVA, sorbitol, and magnesium stearate appeared with minor changes. These changes can be attributed to the formation of hydrogen bonds among the hydroxyl groups of tacrolimus, PVA, and sorbitol. Tacrolimus was dissolved in an ethanolic mixture of HPMC to extrude it through the PAM using a second nozzle to prevent drug decomposition under the printing temperature. Obviously, the DSC results indicated that the crystallinity of the tacrolimus remained intact during the printing process. Furthermore, no decomposition occurred under the printing temperature according to the results obtained from the TGA experiment.

However, our tablet’s results differed in crystallinity from those of tacrolimus in a disintegrating tablet which was developed by Ponnammal et al. through the hot-melt extrusion of a drug and polymer mixture (HPC, Soluplus, and HPMC) at a temperature of about 135 °C, converting to an amorphous form [25]. In contrast, for our 3D-printed tablet, no temperature treatment was applied during printing through extrusion-based PAM, ensuring the stability of the API during the production process.

In the study, we have shown that the 3D-printed tablet can meet the routine physiochemical characterization of traditional tablets in pharmaceutical companies according to USP requirements (assay, content uniformity, and dissolution tests) as well as intra-process quality control, including weight variation, hardness, and friability tests.

According to the dissolution profiles, an extended-release manner for the 3D-printed tacrolimus tablets was found compared to the burst-release trend of commercial tacrolimus capsules. One of the regulatory requirements in developing pharmaceutical dosage forms is establishing the similarity between the dissolution profile of a generic drug and its brand by the similarity factor, f2. Interestingly, a similarity value of more than 60% was statistically obtained between the dissolution profiles of Envarsus XR and 3D-printed tablets, indicating successful formulation of the sustained-release 3D-printed tablet of tacrolimus. Furthermore, our observations showed that the drug release obeys the infill percentage, which is considered for the 3D-printing process. The release profiles were infill%-dependent. The tablet with an infill of 90% had a desired release profile. The 3D-printed tablets were considered stable under accelerated conditions.

Some extended-release dosage forms of tacrolimus, such as Advagraf^©^ and Astagraf XL^©^ tablets, have been developed to simplify the dosing frequency as well as patient compliance for heart, liver, lung and kidney transplantations [26]. Furthermore, a particular extended-release tacrolimus tablet was formulated by the melt-based solid dispersion of tacrolimus and direct compression after blending it with the hydrogel matrix of HPMC. The drug in solid dispersion was in a molecular or amorphous state. The drug release profiles depended on the HPMC type and its content as well as the main lipid materials used in the solid dispersion [27]. The only study that used 3D-printing technology to create a dosage form containing tacrolimus involved a 3D-printed suppository for ulcerative colitis treatment, which was developed by Seoane-Viano et al. Different ratios of the excipients, coconut oil and semi-solid Gelucire, were melted under a temperature below 48 °C and transferred into semi-solid extrusion-based 3D printing. In comparison, the drug converted to an amorphous state and more than 80% of the tacrolimus was released from the suppository after 2 h [28].

Taken together, the extended-release form of tacrolimus tablets developed by an extrusion-based 3D-printing technique in our study conforms to all the obligatory pharmaceutical tests while being economical, versatile, controllable, and relatively effortless to fabricate. The polymers used in our study are also user-friendly, as they can manipulate themselves to load the required amount of drug seamlessly while avoiding thermal degradation throughout the production process and controlling the desired drug release kinetically. The 3D-printed tablet has a high hardness value with no friability, making it suitable for downstream large-scale pharmaceutical processes such as coating, packaging, and handling. Another major advantage of 3D-printing-based manufacturing would be its ability to precisely control dose adjustment, integrate multiple drugs into a pill, and modulate the dissolution rate using different polymer mixtures and infill percentages. In addition, the rapid production technique can provide tailored dosages for point-of-care patients in the personalized medicine field. Specially, the fabrication of the extended-release tablet of tacrolimus using 3D printing in a desired dose would help clinicians to personalize treatment and tacrolimus TDM after transplantation [29].

## 5. Conclusions

In the present study, for the first time, we have developed a novel 3D-printed tablet of tacrolimus in a stable extended-release form using HME and PAM-based 3D printing to provide a dosage for preventing graft rejection in the organ transplantation procedure. No drug degradation nor polymer decomposition were observed during the printing process. The kinetic release of the tablet, which is modulated using different polymer mixtures and infill percentages, can enhance patient compliance with one medication per day. The developed tablet met all the acceptance values of USP requirements upon assay, content uniformity, dissolution tests, and microbial tests as well as the features of intra-process quality control. The feasible, economical, and controllable 3D-printing process can bypass the expensive manufacturing steps in pharmaceutical companies and quickly provide a tailored tacrolimus tablet for personalized medicine at the point-of-care.

## Figures and Tables

**Figure 1 materials-16-04927-f001:**
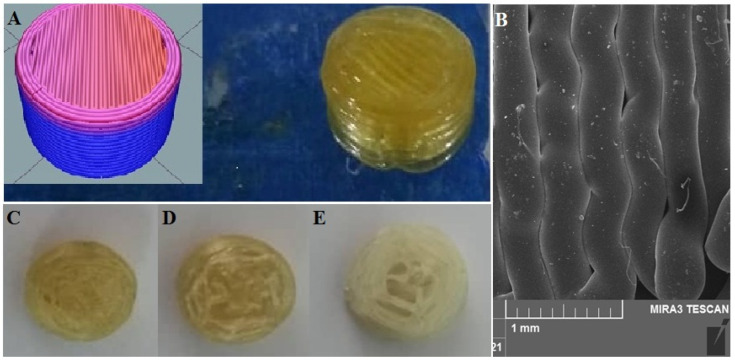
Representative images of 3D-printed pill of tacrolimus (**A**), inset shows the sliced version by software; SEM micrographs of printed layers of PVA filaments (**B**); cross-section of 3D-printed pill with (**C**) 90% infill; (**D**) 50% infill; and (**E**) 10% infill.

**Figure 2 materials-16-04927-f002:**
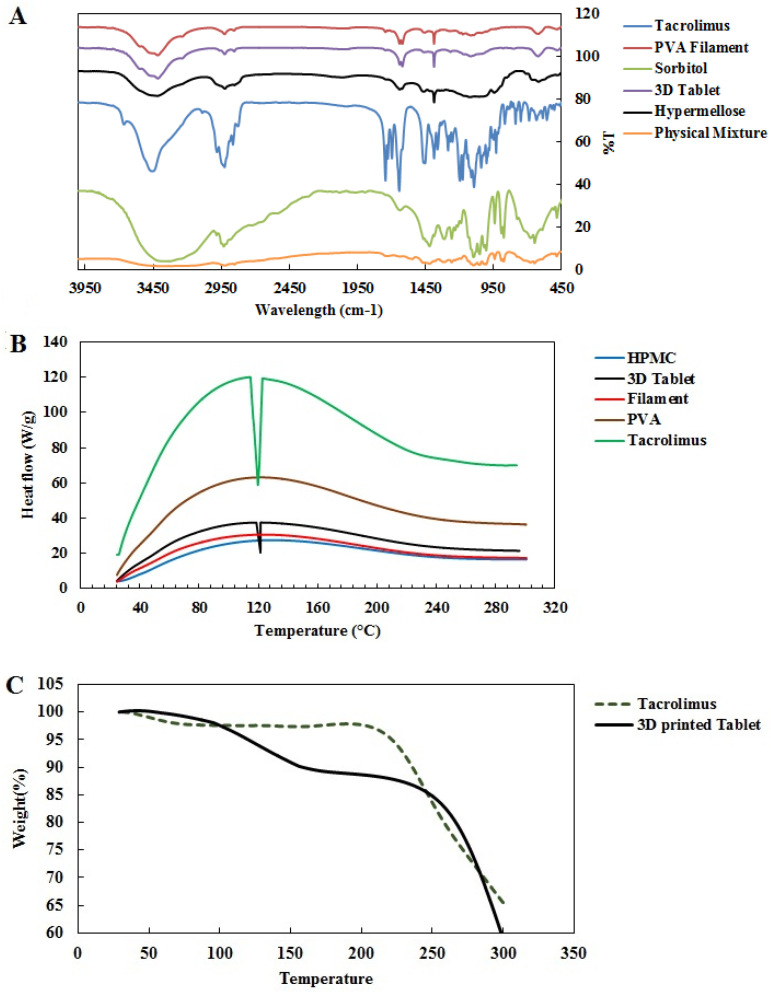
FTIR spectrum of 3D-printed pills (**A**); DSC thermogram of 3D-printed tacrolimus pills and their ingredients (**B**); thermogravimetric analysis of tacrolimus monohydrate and 3D-printed tablet (**C**).

**Figure 3 materials-16-04927-f003:**
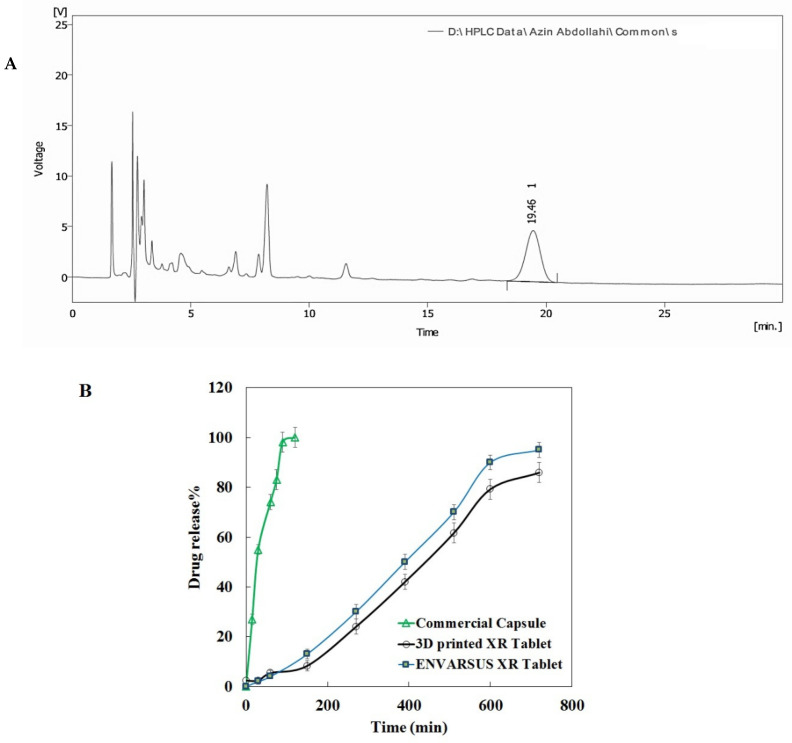
Representative HPLC chromatogram of injected samples of assay solution (**A**); dissolution profile of 3D-printed tacrolimus of 5 mg in comparison with “ENVARSUS” XR tablet and commercial tacrolimus capsule (**B**).

**Table 1 materials-16-04927-t001:** The ratio of components in filament formulation.

Formulation *	F1	F2	F3	F4	F5	F6	F7	F8	F8	F9
PVA filament	7950	7800	6900	6000	6900	6000	7350	6900	6900	6000
Sorbitol	0	0	0	0	0	0	450	900	900	1800
PEG 10,000	0	0	900	1800	0	0	0	0	0	0
PEG 35,000	0	0	0	0	900	1800	0	0	0	0
Mg Stearate	0	150	150	150	150	150	150	150	150	150
Temperature (°C)	>225	>225	>220	×	×	×	195	185	185	150
Printability	✓	✓✓	×	×	×	×	✓✓✓	✓✓✓✓	✓✓✓✓	✓✓✓✓✓
Extrudability	✓	✓	×	×	×	×	✓	✓	✓	✓

* All materials are added in mg.

**Table 2 materials-16-04927-t002:** The report of stability study for 3D-printed tablets of tacrolimus (short-term, accelerated condition).

Stability Specification						
Row	Test Item	Time (Month)	0	1	3	6
		Standard				
1	Appearance	Yellowish Round Tablet	Complies	Complies	Complies	Complies
2	Hardness	Above 50 kg	Complies	Complies	Complies	Complies
3	Friability	Less Than 1%	Complies	Complies	Complies	Complies
4	Assay	93.0–105.0%	98.9	90.01	98.6	98.21
5	Total microbial count	Less Than 1000 Cfu/g	Less Than 10	Less Than 10	Less Than 10	Less Than 10
	Mold/Yeast count	Less Than 100 Cfu/g	Less Than 10	Less Than 10	Less Than 10	Less Than 10
	*E. coli*	Negative	Negative	Negative	Negative	Negative
	*Salmonella* species	Negative	Negative	Negative	Negative	Negative
	*Pseudomonas aeroginosa*	Negative	Negative	Negative	Negative	Negative
	*Staphylococcus aureous*	Negative	Negative	Negative	Negative	Negative
	*Candida albicans*	Negative	Negative	Negative	Negative	Negative
	*Aspergillus niger*	Negative	Negative	Negative	Negative	Negative

## Data Availability

All data are included in the manuscript and Appendix A.

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
