# Peer review of "Additive Manufacturing of an Extended-Release Tablet of Tacrolimus"

_materials, 2023, doi:10.3390/ma16144927_

Round 1

Reviewer 1 Report

Review report:

Article 1 Additive manufacturing of an extended-release tablet of Tacrolimus

Azin Abdollahi, 1 , Mohammad Akrami2,3* , Ismaeil Haririan2,3,4, Mohammad Irani5 , Mojtaba Kamankesh6 and Emad 4 Ghobadi7 .

Reviewer comments:

General: very interesting manuscript investigating the use of 3D printing for developing a new form of tacrolimus. Innovative approach to overcome possible heat influences on the API with this technique.

1              Good characterization and in general necessary quality control tests of the tablets. However missing are tests on microbial quality of the tablets (also interesting because of water evaporation as seen in TGA analysis) and (plan for) shelf life experiments, at least an elaboration on further development would have been of value for the discussion.

2              General remark: there are already several different forms of tacrolimus available including extended release forms, the authors could emphasize more on the fact that their formulation is extended release and personal (which makes it unique to other personal approaches like liquid formulation). However current forms are available in variety of strength: what is the medical need for the personalized form the authors present in the current manuscript?

3              It is known that tacrolimus knows a high variation in bioavailability and switching between different formulations is not without risk and should be accompanied by therapeutic drug monitoring (TDM). How do the authors see implementation and translation of the developed 3D formulation into the clinic, risk of different bioavailability: TDM necessity, next step bioequivalence study etc etc?

Introduction:

Ln 88-93: some small grammatical errors

Ln 120: Tablet diameter of 12mm is relatively large, dosage is small, maybe the authors can give a motivation for this size and or elaborate if smaller sizes are foreseeable with the used technique

Ln 260, 3.6: 90% was selected, what where results of the evaluation? Table?

Ln 260: , figure 3A: a representative HPLC chromatogram is shown, however information/explanation concerning the peaks at the beginning of the chromatogram is lacking. Moreover information about impurities is missing. (revision of 3.6 is advised)

Ln 325: grammatically incorrect, do the authors mean that their results were different from Ponnammal et al?

 discussion should place the developed formulation more in perspective (also in relation to future development and/or implementation in clinic)

Author Response

Dear reviewer

Please find the attached files containing Point by point responses to your questions.

Best

Dr Akrami

Reviewer 2 Report

The MS is interesting and well written.

I have some minor suggestions with a pure academic spirit.

1. The abstract must be smoothed to better summarize the sections.

2. Please insert a clear purpose.

3. Avoid short paragraphs, as for example the par 2.4

4. Check the resolution of figure 3.

5. Insert the limitations in the discussion.

6. Results are arranged into themes. This is good but please introduce them before by means of a few senetnces.

7. Minimize the use of acronyms

Author Response

Dear honorable reviewer

Best

Dr Akrami

Reviewer 3 Report

This study evaluates potential combined application of two 3D printing techniques in the development of tacrolimus extended release tablets. Introduction of 3D printing in the pharmaceutical field will offer numerous benefits, so this topic is certainly worth investigating. However, I consider that this manuscript is unsuitable for publication in the present form. These are the major flaws:

1. I was unable to perform complete review, since there are no tables and supplementary material available for download and authors cited several tables and supplementary figures in the text.

2. It is not clear what the novelty of this study is. Changing of API is only minor extension of the previous studies.

3. How authors selected optimal formulation composition? There are no data regarding composition of the tested formulations.

4. Why only one printed formulation was finally tested, despite more formulations were printable? It was reasonable to evaluate effects of formulation composition and infill density on formulation properties.

5. FTIR spectra should be analysed with respect to shifting of characteristic absorption bands of API, as an indicator of possible interactions.

6. Language editing of the text is necessary.

Author Response

Dear honorable reviewer

Best

Dr Akrami

Round 2

Reviewer 3 Report

The authors made some minor improvement of the manuscript. Although there is not so much novelty in the study, combination of two 3d printing technologies may be interesting.

Incorporation of microbial testing does not have so much sense, since if fabrication of tablets is done in clean environment with excipients which pose required microbial quality, risk for microbial growth in tablets is low.

There is mistake in stability testing, study was performed during 6 months, not 36 months.

It is not clear how the authors ranked printability of formulation F1, F2 and F7-F9.

Concentration units of HPMC mixtures should be added to the table.

The authors should discuss whether characteristic peaks of API are preserved or shift in the spectra of powdered tablets. It is desired to perform FT-IR analysis of corresponding physical mixtures in parallel.

How the authors assessed similarity between dissolution profiles. Does 60% similarity mean that similarity factor is 60? Please add more details for clarification.

Author Response

Response to the Reviewer comments:

  1. The authors made some minor improvement of the manuscript. Although there is not so much novelty in the study, combination of two 3d printing technologies may be interesting.

Thank you so much for your suggestion. Indeed, the work is a combination of two extrusion-based-3D printing methods: PAM (pressure assisted-micro-syringe) and Melt extrusion in parallel. The combination of two methods is mentioned in the abstract and sections 2.2, 2.3. 3.1, 4 and 5. Accordingly, the combination was also highlighted in the abstract in the revised version.

  1. Incorporation of microbial testing does not have so much sense, since if fabrication of tablets is done in clean environment with excipients which pose required microbial quality, risk for microbial growth in tablets is low.

Thanks for your precise consideration. Your suggestion is correct. Incorporation of the microbial test was, according to the recommendation of another respectful reviewer for this article. Furthermore, as we assessed the 3D printed tablet according to United States pharmacopeia (USP) requirements, USP General Chapters of <61> and <62> strongly recommend doing the microbial test for pharmaceutical dosage forms like tablets. USP <61> is often called a "Bioburden" or "Microbial Limits" test. This test determines how many microorganisms are present in non-sterile drug products. USP 62 is performed in routine to test the presence of specified microorganisms: Staphylococcus aureus, Pseudomonas aeruginosa, Escherichia coli, bile-tolerant Gram-negative bacteria, Clostridia species, Salmonella species and/or Candida.

  1. There is mistake in stability testing, study was performed during 6 months, not 36 months.

Thanks a lot. The stability testing was performed according to ICH guidelines Q1 in accelerated condition (6 month period), which is routine in pharmaceutical research and development. It means testing at a more severe accelerated condition can be an alternative to storage testing.

  1. It is not clear how the authors ranked printability of formulation F1, F2 and F7-F9.

Thanks a lot for your valuable suggestion. It was corrected as mentioned. Indeed, it was ranked according to the temperature of the printing process. The best one (F9) was printed at the lowest temperature. However, working stepper motors was more under pressure for F1 than F2.

  1. Concentration units of HPMC mixtures should be added to the table.

Thanks for the excellent corrective suggestion. It was corrected in the table.

  1. The authors should discuss whether characteristic peaks of API are preserved or shift in the spectra of powdered tablets. It is desired to perform FT-IR analysis of corresponding physical mixtures in parallel.

The FTIR analysis of the physical mixture was performed during the revision time and the results was added and discussed as follow: “The FTIR spectrum of physical mixture of the ingredients showed a broad peak of 3342 cm-1 for hydroxyl stretching and peaks of 2918 cm-1 for vibrations of C-H, 1095 cm-1 and 1048 cm-1 for stretching vibration of C-O bands, 887 cm-1  and 1417 cm-1 for bending vibrations of inward and outward O-H bonds. Peaks with a weak intensity at 1638 cm-1 and 1735 cm-1 are attributed to C=O stretching in ester and ketone bands of Tacrolimus.

“The characteristic peaks of the Tacrolimus in the FTIR spectrum of the drug physical mixtures with PVA, sorbitol and magnesium stearate appeared with minor changes. This change can be attributed to the formation of hydrogen bond among hydroxyl groups of Tacrolimus, PVA and sorbitol.”

  1. How the authors assessed similarity between dissolution profiles. Does 60% similarity mean that similarity factor is 60? Please add more details for clarification.

In today's drug development industry, establishing the similarity of dissolution profiles is a regulatory requirement. According to U.S. Food and Drug Administration (USFDA), similarity factor is defined as the logarithmic reciprocal square root transformation of one plus sum of the squared differences between the dissolution values of test and reference products over all time points.

Two dissolution profiles to be considered similar and bioequivalent, if statistical value of f2 be between 50 and 100.

F2 value of 60% means two dissolution profiles are considered 60% similar and equivalent.

The text was corrected as suggested.
